# *Toxoplasma gondii* Infection in Humans: A Comprehensive Approach Involving the General Population, HIV-Infected Patients and Intermediate-Duration Fever in the Canary Islands, Spain

**DOI:** 10.3390/diagnostics14080809

**Published:** 2024-04-12

**Authors:** Cristina Carranza-Rodríguez, Margarita Bolaños-Rivero, José-Luis Pérez-Arellano

**Affiliations:** 1University of Las Palmas de Gran Canaria, Spain and Research Institute of Biomedical and Health Sciences, 35016 Las Palmas de Gran Canaria, Spain; luis.perez@ulpgc.es; 2Microbiology Division, Complejo Hospitalario Universitario Insular Materno Infantil, 35016 Las Palmas de Gran Canaria, Spain; mbolriv@gobiernodecanarias.org

**Keywords:** *Toxoplasma gondii*, AIDS/HIV, Canary Islands, Spain, human toxoplasmosis, seroprevalence

## Abstract

A prior investigation in 1993 identified a high seroprevalence of toxoplasmosis (63%) in the Canary Islands. This study aims to assess the current prevalence of the disease in diverse population groups. The study was based on a population-scale screening involving 273 residents utilizing *T. gondii* IgG ELISA and a 20 year retrospective study (1998–2018). This included AIDS/HIV outpatients (1357, of which 324 were residents), AIDS/HIV hospitalized patients (741) and patients with fever of intermediate duration (158). The seroprevalence in the resident population was 37%, with significant differences between islands. Among resident outpatients with AIDS/HIV, 14.2% had specific anti-*T. gondii* IgG, and three had anti-*T. gondii* IgM; however, IgG avidity testing indicated non-active infection. In patients hospitalized for AIDS/HIV, *T. gondii* causing encephalitis was detected in 2%. Among patients with fever of intermediate duration, 28.5% were positive for *T. gondii* IgG, and four also showed IgM positivity, although the infection was non-active. The study reveals a decrease in human toxoplasmosis over the past 30 years. However, the current seroprevalence, which stands at 37%, together with the substantial risk that *T. gondii* represents for immunocompromised individuals, highlights the need to implement preventive and control strategies to control the threat that this infection can pose to public health in the Canary Islands population.

## 1. Introduction

Toxoplasmosis is a worldwide zoonotic infection caused by the apicomplexan protozoa of the genus *Toxoplasma* [1,2]. Although previously controversial, it is now accepted that the only pathogenic species of the Toxoplasma genus is *T. gondii* [2,3]. This coccidium has a heteroxenous life cycle consisting of an asexual phase of development in various tissues of intermediate hosts and a sexual phase of development in the gut of definitive hosts (Figure 1A) [3].

There are three infectious stages in the life cycle of *T. gondii*: sporozoites included in sporulated oocysts, tachyzoites (endozoites) and bradyzoites (or cystozoites) contained in tissue cysts (Figure 1B) [1,2]. The generation of oocysts takes place in the gut of definitive hosts. Once eliminated in feces and under appropriate conditions of temperature, humidity and aeration, sporulation occurs within 1 to 21 days [1]. Sporulated oocysts are infective when contaminating soil, water, fresh food (i.e., fresh vegetables, cheese, milk) or bivalve mollusks [4,5,6]. Sporulated oocysts are highly resistant to cold, drying, and multiple disinfectants (i.e., iodide or chloride) [1,3,5,6]. Tachyzoites are smaller than oocysts and have a crescentic or oval shape [1]. They are found in intermediate hosts and are usually derived from the transformation of orally ingested sporozoites or bradyzoites. Tachyzoites have a great capacity to replicate and penetrate all cell types (except erythrocytes). Inside the host cell they are in a parasitophorous vacuole expressing parasite proteins that prevent fusion with lysosomes and thus destruction of the tachyzoites. After several replication cycles, the host cell is disrupted and the tachyzoites are disseminated by blood to the different tissues. The main locations are shown in Figure 2.

In immunocompetent individuals there is a potent T helper 1 response with the production of proinflammatory cytokines (i.e., TNFa, γ IFN, IL-12) that control parasite growth and lead to the generation of tissue cysts consisting of bradyzoites. Exceptionally, tachyzoites can be transmitted to humans by blood through transfusions, biological accidents or in parenteral drug users [3]. Unlike oocysts and tissue cysts, tachyzoites are very sensitive to environmental conditions and do not survive outside the host. Bradyzoites are the main component of tissue cysts and are morphologically similar to tachyzoites, although their growth is very slow, and they can persist throughout the life of the host. In the life cycle of *T. gondii*, bradyzoites play a dual role. On the one hand, they constitute a horizontal transmission route between omnivorous intermediate hosts and, on the other hand, they are responsible for the recrudescence of the infection in immunosuppressed individuals by transforming into tachyzoites. Bradyzoites are more resistant than tachyzoites, including inactivation by digestive enzymes (i.e., pepsin and trypsin). However, they are more sensitive than sporozoites to temperature changes (inactivated by freezing to −12 °C or lower or by heating to 67 °C), meat processing (i.e., curing with salt, or low temperature smoking) or gamma irradiation.

The life cycle of *Toxoplasma gondii* includes two types of ***host***: definitive and intermediate hosts [2,4]. The definitive hosts, in which the sexual phase takes place, are domestic cats as well as multiple species of wild felines. In them, *T. gondii* infection may be due to ingestion of embryonated oocysts or ingestion of tissue cysts from an intermediate host. Although there are differences depending on the form of infection (embryonated oocysts or tissue cysts), in both cases there is a final stage of sexual differentiation leading to the formation of oocysts. It is of interest to note that in definitive hosts, *T. gondii* infection is usually asymptomatic, is not associated with vertical transmission and direct contact with intermediate hosts does not carry a risk of infection, due to the need for sporulation of oocysts in the environment. Multiple intermediate hosts, including humans, are susceptible to infection with this parasite. The main ones are: (i) livestock (pigs, cattle, sheep, goats, poultry, horses), (ii) pets (dogs, canaries, finches), (iii) wild animals (deer, bears, raccoons), (iv) zoo animals (New World primates and kangaroos) and (v) marine mammals (sea otters, seals, sea lions and dolphins). The consequences of *T. gondii* infection differ markedly depending on the species considered. Thus, cattle and horses are considered poor hosts and no confirmed cases of clinical toxoplasmosis or abortion have been reported. In contrast, infection of sheep and goats has been clearly associated with abortion and neonatal mortality.

Human T. gondii infection has two main patterns: primoinfection (through several horizontal or vertical routes) and reactivation. Horizontal transmission forms are [2,5,6,7,8]: (i) consumption of fresh food, water or bivalve mollusks contaminated by sporulated oocysts, (ii) oral ingestion of tissue cysts contained in raw or undercooked meat or viscera of intermediate hosts, (iii) blood tachyzoites inoculated accidentally or in intravenous drug users and (iv) recipients of organ transplants from infected donors (tissue cysts and/or tachyzoites). Epidemiologically, oral transmission (oocysts or tissue cysts) is the most important. Although it has been controversial [4,5], study of well-characterized outbreaks suggests that the severity of toxoplasmosis acquired by oocysts or tissue cysts is similar. Vertical transmission (mother-to-child) is another important route of infection that usually occurs through placental involvement by tachyzoites during pregnancy or in the weeks prior to pregnancy. Other less frequent forms of congenital toxoplasmosis have been described in children of women previously infected with T. gondii in whom immunosuppression (i.e., acquired immunodeficiency syndrome (AIDS) or systemic lupus erythematosus) was present [2]. The risk of vertical transmission is lower in early pregnancy, although its consequences are more severe (i.e., miscarriages). On the contrary, it increases in the last third of pregnancy, and neonates may be apparently normal but with cerebral manifestations (hydrocephalus and calcifications) or ophthalmic manifestations later [9]. After primary infection, a latency phase is established, which may last a lifetime in immunocompetent individuals. However, in the presence of immunosuppression (especially AIDS or transplant recipients), bradyzoites transform into tachyzoites and may cause clinical manifestations by reactivation. Although there are experimental data on the possibility of toxoplasmosis after the use of anti-TNFα [10], we have not found clinical data to support this relationship. Reactivation may be systemic or, more commonly, restricted to the central nervous system, causing encephalitis characterized by fever, headache, ataxia, seizures, decreased level of consciousness and memory loss [11,12]. A specific form of this disease is isolated ocular toxoplasmosis, characterized by the presence of chorioretinitis, with visual disturbances, whose most typical manifestation is the image in the fundus called “headlight in the fog” [13,14]. Isolated ocular toxoplasmosis may be due to either primary infection or reactivation. Among the factors related to the development of ocular toxoplasmosis, the relationship with different geographic areas and the presence of specific *T. gondii* genotypes have been observed [15,16]. Toxoplasma includes the three archetypal genotypes (I, II and III), together with non-archetypal genotypes organized in 16 haplogroups that differ in prevalence, virulence, migratory capacity within the host and different cytokine responses. Specifically, ocular toxoplasmosis has been associated with genotype II and South American origin [5,13]. *Reinfection* by *T. gondii* is exceptional, probably related to the generation of IgA isotype antibodies [1].

In general, the clinical manifestations of toxoplasmosis, except in the previously mentioned situations (congenital, immunodepressed patients or ocular) have been considered as infrequent. However, because of its nonspecific nature (i.e., fever, cervical lymphadenopathy, myalgia, or fatigue) and its limited duration, which limits diagnostic suspicion [5], the real incidence of acute toxoplasmosis as fever of intermediate duration is probably underestimated.

Diagnosis of T. gondii infection can be made directly [by PCR (blood, cerebrospinal fluid, vitreous or aqueous humor and urine), immunohistochemistry or histology] or indirectly by serological methods (see below).

The epidemiology of T. gondii infection worldwide is highly variable depending on geographical location, changes in habits and trends within the population (i.e., linked to the consumption of fresh vegetables), the control of some forms of immunosuppression (i.e., HIV) or the early detection of congenital infection [6,17,18].

Thus, the aim of this study was to assess the prevalence of toxoplasmosis within distinct population groups among the residents of the Canary Islands.

## 2. Population and Methods

This study included two groups of individuals: the general population of the Canary Islands and two types of patients evaluated by a retrospective review of medical records between January 1998 and December 2018 at the Hospital Universitario Insular, a tertiary referral hospital located in Gran Canaria, Spain (Figure 3).

### 2.1. Population-Based Survey

The presence of IgG antibodies against *Toxoplasma gondii* was determined by quantitative enzyme immunoassay in the serum of 273 individuals representing the population of the seven Canary Islands.

This sample size was calculated based on a total population of 2,000,000 people, an expected frequency of 30%, and an acceptable margin of error of 5.5% (*n* = 267). Thus, 273 residents were randomly selected, with 30/31 individuals chosen from islands with a low population (<150,000 inhabitants; 5 islands) and 60 people from islands with a high population (>150,000 inhabitants; 2 islands). Variables such as age (young: <18 years old; adult: 18–64 years old; elder: ≥65 years old) and gender were included in the study. Informed consent was obtained from all individual participants enrolled in the study.

### 2.2. HIV-Infected Patients

Adults aged over 18 years with confirmed HIV infection, both born and residing in the Canary Islands, were included in the study. Comprehensive data were collected through the standardized review of medical records, encompassing demographic characteristics such as age, gender, clinical and laboratory data and outcome related to toxoplasmosis.

A total of 1357 HIV-infected patients with toxoplasma serology data were examined, and among them, 324 individuals were confirmed to be both born and residing in the Canary Islands. Notably, the remaining 1033 patients were identified as foreign individuals, originating from other regions of Spain, Europe or with unclearly established origins or residence.

Additionally, we conducted a review of 741 hospitalized patients with HIV infection during the study period, specifically focusing on those presenting with focal brain lesions attributed to *Toxoplasma gondii.*

### 2.3. Fever of Intermediate Duration (FDI)

FID is defined as a non-localized fever occurring in the community, lasting between 1 and 3 weeks, and lacking diagnostic orientation after standard clinical, analytical, and radiological evaluations are completed [19,20,21]. A total of 158 adult patients over 18 years with FDI who were both born and residing in the Canary Islands were reviewed during the study period. Routine serological testing for *Toxoplasma gondii*, along with other pathogenic microorganisms, was performed for diagnostic purposes in these patients.

### 2.4. Serological Study

For the population-based survey, specimens were collected in sterile tubes without anticoagulant, allowed to clot, centrifuged and then the serum was separated. Sera were aliquoted and kept frozen at −20 °C until the time of testing. The ELISA kit NovaLisa^®^ *Toxoplama gondii* IgG ELISA (Mikrogen Diagnostik, Neuried, Deutschland) was used to detect the serum for IgG and IgM class antibodies following the manufacturer’s recommendations. The microtiter strips were precoated with inactivated *T. gondii* antigen. Serum samples were diluted 1:100 and added to the wells, incubated and washed. HRP-labeled anti-human IgG conjugate was added, further incubated and then plate washed. TMB substrate was added followed by stop solution and OD readings were taken. Positive and negative controls were included per batch of test run to ensure that reagents were working properly, and technical procedures were carried out correctly. The sensitivity and the specificity of the kit was 96.6% and 98.2%, respectively. The interpretation of the results was as follows: reactive, >35 IU/mL; grey zone (equivocal), 30–35 IU/mL; non-reactive, < 30IU/mL.

For HIV and FID, all commercial tests were performed at the Microbiology Unit of the Hospital Universitario Insular de Gran Canaria (Spain) and were interpreted according to the manufacturer’s instructions.

Toxo IgG II (VIDAS-IgG^®^) and Toxo IgM (VIDAS-IgM^®^) tests (bioMérieux, Marcy l’Étoile, France) are fully automated ELISAs for the quantitative determination of IgG and IgM antibodies against *T. gondii* in human serum using microparticles coated with native SAG1 antigen [22].

The IgG avidity assay has emerged as a crucial tool for distinguishing between acute and chronic toxoplasmosis [23]. Since anti-toxoplasma IgM antibodies may persist for extended periods post-infection, their sole presence is insufficient for diagnosing primary infection. The same holds true for anti-toxoplasma IgG antibodies, which manifest within two weeks of primary infection and remain detectable throughout life [2]. Differentiating acute from chronic infection can be achieved through IgG “avidity” testing. This method relies on the observation that prolonged immunologic exposure to the organism results in the production of anti-toxoplasma IgG antibodies with progressively stronger binding (avidity) to toxoplasmal antigens. Therefore, in a patient with positive IgM, weaker binding of IgG in an avidity assay suggests a more recent infection [24].

Originally developed by Hedman et al. in Finland [25], this method is based on the dissociation of hydrogen bonds between antigens and antibodies using urea [26]. A high avidity value (AI ≥ 60%) indicates that toxoplasma infection occurred more than 3 months ago. Conversely, borderline avidity (40% < AI <60%) suggests infection over an undetermined period, while low avidity (AI ≤ 40%) indicates infection within the last 3 months. The VIDAS Toxo IgG II Avidity test (bioMérieux, Marcy l’Etoile, France) was employed for this purpose.

### 2.5. Statistical Analysis

Data were entered in a database and descriptive statistical analysis was performed using Epi-InfoTM, version 7.2, offered in open by CDC, Atlanta (https://www.cdc.gov/epiinfo/index.html (accessed on 9 May 2019) Associated risk factors were evaluated by StatCal 2 × 2 tables, by which Odds Ratio (OR), Risk Ratio (RR), Chi2, Confidence interval –CI- (95%), Mantel–Haenszel (M–H) and *p* value were obtained. *p* values were considered as significant when *p* < 0.05.

## 3. Results

The seroprevalence in the overall population of the Canary Islands was 37% (101/273), with statistically significant differences between islands (Table 1).

The highest seroprevalence (17/31; 54.8%) was found in La Palma (OR = 2.2840; 95%CI: 1.0732–4.8608; *p* = 0.0171), while the lowest (4/30; 13.3%) was detected in Lanzarote (OR = 0.2316; 95% CI: 0.0784–0.6843; *p* = 0.0015) (Figure 4). Odds Ratio, as an outcome variable, was >1 in all four west islands (La Palma, El Hierro, La Gomera and Tenerife); in contrast, OR < 1 was observed in all three east islands (Gran Canaria, Fuerteventura and Lanzarote). Relative risk, as a predictor variable, was >1 in the west islands and <1 in the east islands.

Concerning the age of the surveyed people (Table 2), the highest seroprevalence (47/62; 75.8%) was observed in older people (OR = 9.11; 95% CI: 4.71–17.06; *p* ≤ 0.001), while the lowest (10/97; 10.3%) was found in the young population (OR = 0.11; 95% CI: 0.05–0.22; *p* < 0.001). Adult people, in between, showed a seroprevalence of 38.6% (44/114) (OR = 1.12; 95% CI: 0.68–1.85) although differences were not statistically significant (*p* = 0.3223). Regarding gender (Table 2), seroprevalence in females was 34.7% (51/147) (OR = 0.81; IC 95%: 0.49–1.32), while in males it was 39.7% (50/126) (OR = 1.24; 95% CI: 0.76–2.03); nevertheless, differences were not statistically significant (*p* = 0.12).

In asymptomatic HIV-infected patients born and residing in the Canary Islands, the seroprevalence of *T. gondii* was 46/324 (14.2%), in 3 of which anti-*T. gondii* IgM was also detected. IgG avidity testing performed in these IgG + and IgM + patients revealed inactive *T. gondii* infections (>60%). Among the seropositive study participants, 36 (80%) were men. More than 51.1% of the HIV-positive people were less than 31 to 45 years old. The mean age of HIV patients infected with *Toxoplasma gondii* increased during the study period, from 35 years (SD 13) in the first decade (1999–2008) to 43 years (SD 10) in the second decade (2009–2017) (*p* < 0.031). Thirty-five (76%) of the seropositive study participants had a CD4 + lymphocyte count of ≥200 cells/µL) and the mean CD4+ lymphocyte count was 423 ± 280 cells/µL (Table 3).

Regarding hospitalized HIV-infected patients, fifteen HIV/AIDS patients (2%) had cerebral toxoplasmosis from among all 741 HIV/AIDS patients admitted between January 1998 and December 2017 and who were part of this retrospective observational study. In addition to affecting the central nervous system, in one patient it also affected other organs (in a disseminated form), and in another patient the parasite was detected infecting the lungs (a pulmonary form). The number of admissions in HIV-infected patients with the *Toxoplasma gondii* decreased by 72%, from 12 patients in the first decade (1999–2008) to 3 in the second decade (2009–2017). The presence of other AIDS-defining conditions in addition to *Toxoplasma gondii* infection was common, particularly bacterial pneumonia (20%), tuberculosis (13.3%) and esophageal candidiasis (6.6%). Coinfection with hepatitis C virus (13.3%) was also frequent. Table 4 summarizes the clinical and demographic characteristics of the fifteen patients.

With respect to patients with fever of intermediate duration, of 158 assessed individuals, 45 showed *T. gondii* IgG +, while 113 were seronegative. Four of forty-five *T. gondii* IgG + patients also showed IgM+, but the IgG avidity test demonstrated that infection was not active in those patients (> 60%). The number of males was 31 (68.9%), the median age was 49 years (range 26–77). Among 88 study participants who had contact with dogs and/or cats, 26 (29.5%) were seropositive for *T. gondii*.

## 4. Discussion

Seroprevalence, as determined by specific anti-toxoplasma IgG antibodies, provides a measure of cumulative *T. gondii* exposure throughout an individual’s lifetime. Toxoplasmosis has been considered the most prevalent human infection in the world with seroprevalence ranging widely from 0% to 100% [2] and mean values suggesting that between 30 and 50% of the global population may be affected [16]. This wide variability depends on multiple, often interrelated factors: (i) Geographic location. Thus, in the United States, for example, estimates suggest that approximately 11% of the population, equivalent to some 30 million individuals, has been infected by T. gondii while in Europe, seroprevalence ranges from 10% (Norway) to 60% (Germany), in Latin America it ranges from 38% (Venezuela) to 76% (Costa Rica) and in Africa it is very high (i.e., 84% in Madagascar) [16,27]. (ii) Time period of study. In general, there is a worldwide pattern indicating a decrease in the prevalence and incidence of the disease in humans [9,28]. In addition, some authors, in study of toxoplasmosis outbreaks, suggest that transmission patterns and sources of infection vary by decade [18]. For example, in the 1960s and 1970s, ingestion of cysts in meat and meat products was considered the main source of infection. In the 1980s, contamination of milk with tachyzoites was prevalent, whereas in the 2000s, outbreaks were more related to the presence of oocysts in water, sand and soil. Since 2010, the focus has been on oocysts present in raw fruits and vegetables. (iii) Population studied. Most of the seroprevalence data are derived from three groups of people: women of childbearing age, HIV-infected patients and outbreaks of toxoplasmosis. Thus, seroprevalence studies in *pregnant women* conducted in European countries range from 9% in the United Kingdom [29], 31% in Austria [30], to 55.8% in Estonia [31]. Specifically, the Annual Epidemiological Report on Congenital Toxoplasmosis for 2015 in European countries indicated a total of 133 confirmed cases, of which 83% were reported in France due to active screening of pregnant women [32]. On the other hand, prior to the development of highly active antiretroviral therapy (HAART), severe encephalitis caused by *T. gondii* was estimated to affect up to 40% of *HIV* patients worldwide [33]. The capacity of *T. gondii* to persist in the CNS of immunocompetent individuals without causing symptoms is highly unusual and contrasts with most pathogens, which typically cause neurological symptoms upon breaching the blood–brain barrier, often leading to fatal diseases [34]. *T. gondii*’s ability to remain asymptomatic in the CNS of immunocompetent individuals is noteworthy. However, in immunosuppressed patients, *T. gondii* could cause severe neurological disease. With the use of HAART, the incidence of central nervous system (CNS) toxoplasmosis in AIDS patients has consistently declined, and reactivation of latent infection can be prevented through proper prophylaxis [2]. Therefore, the current low prevalence of CNS toxoplasmosis in these patients underscores the efficacy of therapeutic and preventive measures. Finally, the study of toxoplasmosis outbreaks has provided important information about the factors involved in its pathogenesis and epidemiology [5,35]. In relation to the aforementioned factors, there are multiple associated data (demographic, environmental and socioeconomics) such as age, rural or urban habitat, profession, health practices, dietary habits, soil moisture and prevalence in intermediate hosts [2,13,16].

Another important factor related to seroprevalence is the technique used in the detection of antibodies [2,36]. The Sabin–Feldman test is considered the gold standard for the serological diagnosis of toxoplasmosis due to its high sensitivity and specificity. However, it is labor-intensive and is currently used to a very limited extent, so other techniques such as enzyme-linked immunosorbent assay (ELISA), microparticle enzyme immunoassay (MEIA), indirect fluorescence antibody test (IFAT), immunosorbent agglutination assays (ISAGA) or hemagglutination are more used. These tests differ in their sensitivity and specificity and may give discordant results.

The overall data on Toxoplasma gondii infection in Spain indicate a highly variable seroprevalence, ranging from 42.4% to 35.1% in the general population [37]. To our knowledge, only one small study in Spain has examined the genotypes associated with *T. gondii*. The findings suggest that genotype 1 is predominant in immunocompromised patients, while genotype 2 is more commonly implicated in congenital toxoplasmosis [38]. It should be noted that serological data have been obtained mainly from studies in pregnant females [39,40,41,42,43,44,45] and, occasionally, in hospitalized patients (with or without HIV infection) [46,47]. Therefore, these data cannot be directly extrapolated to the general population of the country (which also includes people of different age and sex). Seroprevalence rate of Toxoplasma gondii ranges from 5.2% to 51.0% depending on several factors: (i) sample size, (ii) geographical region, (iii) time period of study and (iv) origin of people (autochthonous or immigrants).

In the Canary Islands, an insular region of Spain, data on Toxoplasma gondii infection are very scarce. A previous study conducted among residents of Gran Canaria indicated a high seroprevalence of 63% [48]. Nevertheless, studies conducted on livestock in the same island revealed a notable decrease in toxoplasmosis seroprevalence over time. Specifically, the prevalence declined from 68% in 1995 to 9.8% in 2017, showing an overall prevalence of 7.8% for goats, the predominant livestock species in the Canary Islands [49]. The authors attribute this decline to advancements in animal production practices at the farm level, mirroring a global trend in reducing toxoplasmosis prevalence in livestock.

The overall seroprevalence of 37% in the Canary Islands would be within the prevalence range of European countries and regions. However, it varied according to the island of residence, ranging from 13.3% in Lanzarote to 54.8% in La Palma (Figure 4), approaching the upper thresholds in Europe. According to the Köppen–Geiger climate classification, prevalence is lower in the eastern islands with a BWh climate (hot desert) and higher in the western islands with a Csb climate (temperate). Consequently, statistically significant high prevalence was observed in La Palma and Tenerife (western islands), while statistically significant low prevalence was identified in Lanzarote and Fuerteventura. These findings can be attributed to the favorable conditions of warm and humid climates, promoting the survival and viability of oocysts, which would facilitate the transmission.

Regarding age, substantial differences were evident, with a seroprevalence of 75.8% in older individuals (*p* ≤ 0.001) and 10.3% in young people (*p* ≤ 0.001). Consistent with much epidemiological research indicating an age-related increase in seroprevalence, older individuals have had more exposure opportunities to oocysts. In terms of gender, some studies suggest a higher seroprevalence in males [50], a trend also observed in our study (39.7% vs. 34.7%), although without statistical significance (*p* = 0.19). The limited sample size in this survey may have hindered the detection of statistically significant differences.

Regarding AIDS/HIV outpatients attended at the hospital from 1998 to 2018, 46 out of 324 (16.5%) showed specific antibody titers without active infection, as confirmed by the IgG avidity test. Vulnerable populations, such as HIV patients, often demonstrate higher seroprevalence compared to immunocompetent individuals, where the immune system effectively controls the parasite. Nevertheless, our study revealed that resident HIV patients have shown lower seroprevalence than the general population in Gran Canaria (16.5% vs. 31.7%) and the overall Archipelago (16.5% vs. 37%). These results differ from those provided by a recent meta-analysis indicating a higher worldwide seroprevalence of *T. gondii* in AIDS/HIV patients compared to healthy control (46.12% vs. 36.56%) [51]. The lower seroprevalences observed in patients residing in the three eastern islands could explain these findings. Concerning hospitalized HIV-infected patients, 15 out of 741 (2%) showed CNS lesions associated with *T. gondii* infection.

In patients with intermediate fever (158), 45 showed *T. gondii* IgG +, and among them, 4 also showed IgM+. However, subsequent IgG avidity tests demonstrated no active infection in these patients, thereby excluding *Toxoplasma gondii* as a causative or concomitant agent in all investigated cases of intermediate fever. These results are similar to those reported by our group [52] and other Spanish and international publications [53,54] in which the role of *Toxoplasma gondii* in this syndrome is exceptional.

In conclusion, this study suggests a decrease in human infection by *Toxoplasma gondii* in the Canary Islands in the last 25 years and its minimal importance as a causative agent of intermediate-duration fever. However, the current seroprevalence of 37%, the infection in young people and the substantial risk posed by *T. gondii* to immunocompromised patients highlight the need for further hygienic–sanitary measures “from stable to table”.

## Figures and Tables

**Figure 1 diagnostics-14-00809-f001:**
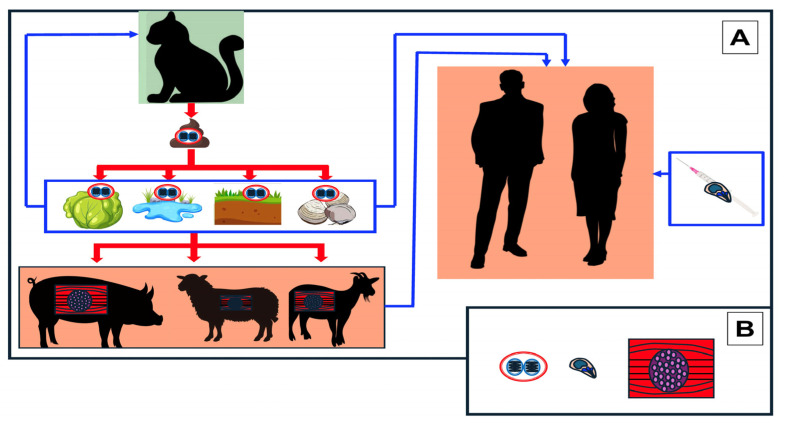
The life cycle of *Toxoplasma gondii.* (**A**) asexual phase of the parasite in intermediate hosts; (**B**) infective parasitic stages:sporozoites, tachyzoites and bradyzoites.

**Figure 2 diagnostics-14-00809-f002:**
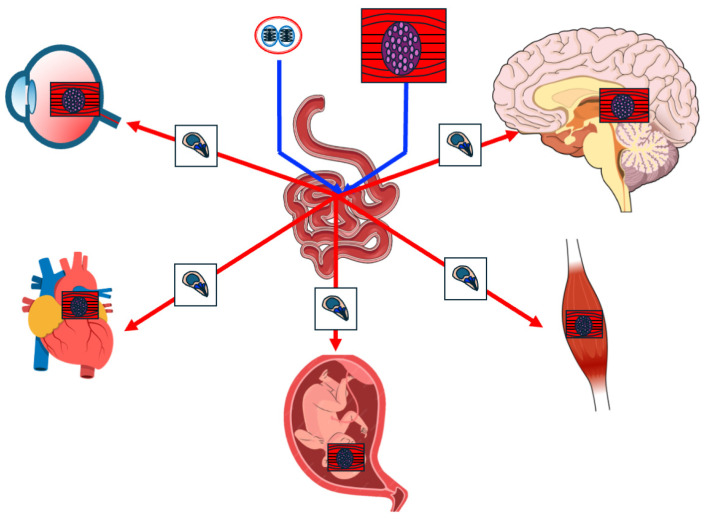
Dissemination of *Toxoplasma gondii* tachyzoites to various tissues: a visual representation.

**Figure 3 diagnostics-14-00809-f003:**
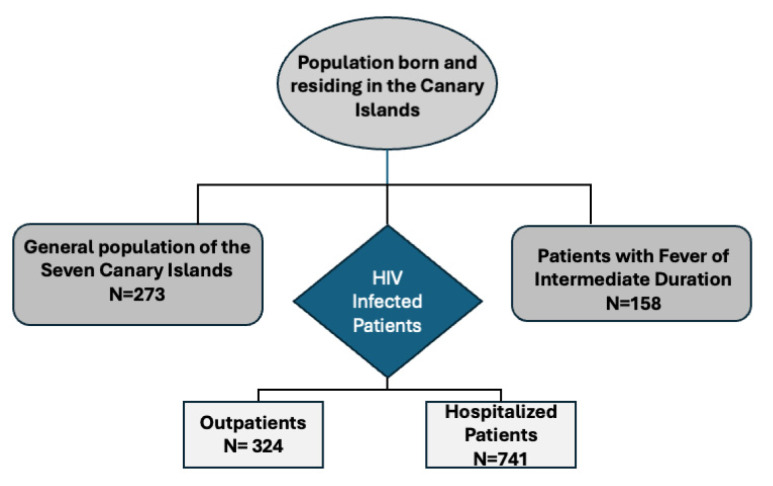
Flow chart of the study population.

**Figure 4 diagnostics-14-00809-f004:**
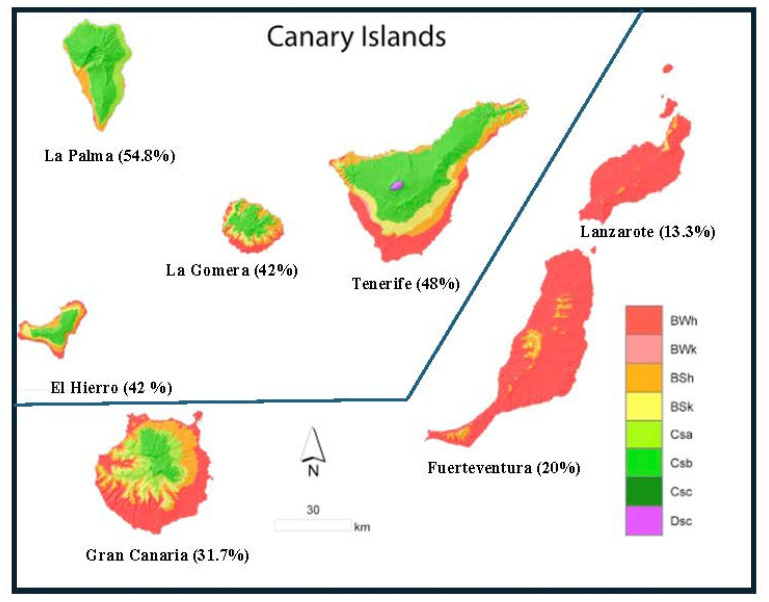
Seroprevalence of *Toxoplasma gondii* in the Canary Islands.

**Table 1 diagnostics-14-00809-t001:** *Toxoplasma gondii* seroprevalence in the Canary population.

Island	N	Pos (%)	OR *	95% CI *	RR *	*p* Value
El Hierro	31	13 (42)	1.2639	0.5911–2.7024	1.1532	0.273
La Palma	31	17 (54.8)	2.2840	1.0732–4.8608	1.5799	0.017
La Gomera	31	13 (42)	1.2639	0.5911–2.7024	1.1532	0.273
Tenerife	60	29 (48)	1.8320	1.0252–3.2737	1.4299	0.021
Gran Canaria	60	19 (31.7)	0.7403	0.4023–1.3625	0.8226	0.169
Fuerteventura	30	6 (20)	0.3895	0.1535–0.9881	0.5116	0.019
Lanzarote	30	4 (13.3)	0.2316	0.0784–0.6843	0.3340	0.001

* OR: Odds Ratio; CI: Confidence Interval; RR: Risk Ratio.

**Table 2 diagnostics-14-00809-t002:** Statistical analysis of age and gender variables in the population of the Canary Islands.

Variable	n	Pos (%)	OR *	95% CI *	RR *	*p* Value
Age						
Young	97	10 (10.3)	0.1074	0.0524–0.2202	0.1994	0.001
Adult	114	44 (38.6)	1.1248	0.6841–1.8494	1.0766	0.322
Elder	62	47 (75.8)	9.1099	4.7161–1,705,970	2.9621	0.001
Gender						
Female	147	51 (34.7)	0.8075	0.4934–1.3216	0.8743	0.199
Male	126	50 (39.7)	1.2384	0.7566–2.0268	1.1438	

* OR: Odds Ratio; CI: Confidence Interval; RR: Risk Ratio.

**Table 3 diagnostics-14-00809-t003:** Characteristics of HIV-infected patients in outpatient consultations.

	IgG +Toxoplasma HIV-Infected Patients *n* = 46	IgG -Toxoplasma HIV-Infected Patients *n* = 278	*p* Value
Sex (n,%)			
Male	36 (80)	253 (91)	0.026
Female	9 (20)	25 (9)	
Age (years) median ± DS *	40 ± 11.9	31 ± 11.4	0.001
Age groups (years) (n,%)			
≤30	9 (20)	120 (45.8)	0.03
31–50	30 (66.7)	126 (48.1)	
>50	6 (13.3)	16 (6.1)	

* DS: Standard deviation.

**Table 4 diagnostics-14-00809-t004:** Characteristics and outcomes of 15 HIV/AIDS patients hospitalized with a complaint of cerebral toxoplasmosis.

Sex (n,%)	
Male	12 (80)
Female	3 (20)
Age (years), median ± DS *	40 ± 11.7
Age groups (years), (n,%)	
≤30	3 (20)
31–50	9 (60)
>50	3 (20)
Diagnosis of HIV infection after current admission (n,%)	8 (53.3)
CD4+ count (cells/mm^3^) median (IQR *)	45 (10–112)
<50	8 (53.3)
50–200	5 (33.3)
>200	1 (6.7)
Viral load (copies/µL) median (IQR *)	152,000 (114,720–512,500)
On ART (n,%)	6 (40%)
Hospital stay (days) median (IQR *)	18 (14–42)
Mortality per year	3 (20%)

* DS: Standard deviation; IQR: Interquartile range.

## Data Availability

C. Carranza-Rodríguez has full access to and is the guarantor for the data. The datasets generated are available from the corresponding author on reasonable request.

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
