# Peer review of "Toxoplasma gondii Infection in Humans: A Comprehensive Approach Involving the General Population, HIV-Infected Patients and Intermediate-Duration Fever in the Canary Islands, Spain"

_diagnostics, 2024, doi:10.3390/diagnostics14080809_

Round 1

Reviewer 1 Report

Comments and Suggestions for Authors

Dear Authors,

Please pay attention to the following questions and comments, pertaining to your manuscript:

1.      Title. Please correct as such: HIV Infected Patients

2.      Please extensively revise your manuscript for structural and grammatical errors. The punctuation should be also revised carefully. Italics should be used ONLY for the species and NOT for the different infectious stages of the parasite or the host types.

3.      Figure 2. Please improve the quality of the figure, it appears very blurry.

4.      Please provide a graphical presentation (flow chart) of your study population. It would be very helpful to have an overview of the three branches (residents, HIV infected out-and inpatients, FDI patients).

5.      Please try to keep the same writing style (font) throughout your text and in the tables.

Best Regards

Comments on the Quality of English Language

Extensive revision

Author Response

  1. Title. Please correct as such: HIV Infected Patients

RESPONSE:

The title has been corrected

  1. Please extensively revise your manuscript for structural and grammatical errors. The punctuation should be also revised carefully. Italics should be used ONLY for the species and NOT for the different infectious stages of the parasite or the host types.

RESPONSE:

The text has been revised and corrected in the manuscript.

  1. Figure 2. Please improve the quality of the figure, it appears very blurry.

RESPONSE:

We send again figure 2 with higher quality.

  1. Please provide a graphical presentation (flow chart) of your study population. It would be very helpful to have an overview of the three branches (residents, HIV infected out-and inpatients, FDI patients).

RESPONSE:

We attach the flow chart and include it in the manuscript as figure 3.

The previous figure 3 is now figure 4.

  1. Please try to keep the same writing style (font) throughout your text and in the tables.

RESPONSE:

The text has been reviewed and corrected in the manuscript.

Reviewer 2 Report

Comments and Suggestions for Authors

The authors evaluated the seroprevalence of T. gondii infection in different populations in the Canary Islands. This is an interesting study that provides important information about a decrease in human infection with T. gondii in the Canary Islands. I have some considerations concerning the manuscript.

The introduction provides a lot of general information about “toxoplasmosis.” I suggest reducing the content and including information concerning the prevalence of toxoplasma infection in the population studied (AIDS/HIV patients, patients with fever of intermediate duration, and the general population) worldwide, in Europe, and in Spain.

Please add captions to each one of the figures.

In page 3. Line 65, please add the symbol “γ”

On page 11, lines 322-326, please revise because some ideas seem contradictory. In lines 344-345, please provide more information concerning the highly variable prevalence. In lines 345-347, please revise because the reference indicates the opposite of the author's description.

Comments on the Quality of English Language

 Minor editing of English language required

Author Response

Reviewer #2: Reviewer's report

  1. The authors evaluated the seroprevalence of T. gondiiinfection in different populations in the Canary Islands. This is an interesting study that provides important information about a decrease in human infection with T. gondii in the Canary Islands. I have some considerations concerning the manuscript.

The introduction provides a lot of general information about “toxoplasmosis.” I suggest reducing the content and including information concerning the prevalence of toxoplasma infection in the population studied (AIDS/HIV patients, patients with fever of intermediate duration, and the general population) worldwide, in Europe, and in Spain.

RESPONSE:

Regarding this comment, we should point out that the initial version of the manuscript included less information but the Editorial Office indicated that the text should have at least 4,000 words, so the information was expanded.

On the other hand, information on INFECTION, as opposed to Toxoplasma gondii DISEASE in patients with HIV is scarce in all geographical contexts since the use of serology is of no practical use and, in fact, is not included in any of the main international guidelines for the evaluation of HIV-infected patients. Finally, data in patients with intermediate duration fever are very scarce and have already been included in the discussion.

  1. Please add captions to each one of the figures

RESPONSE:

Revised and added figure captions

  1. In page 3. Line 65, please add the symbol “γ

RESPONSE:

Added the symbol in the manuscript.

  1. On page 11, lines 322-326, please revise because some ideas seem contradictory. In lines 344-345, please provide more information concerning the highly variable prevalence. In lines 345-347, please revise because the reference indicates the opposite of the author's description

RESPONSE:

All suggestions have been reviewed and clarified in the text.

Round 2

Reviewer 1 Report

Comments and Suggestions for Authors

Dear Authors,

thank you for providing comprehensive and convincing answers to the questions and queries expressed by me and the other Reviewers and made changes, which have contributed to the optimization of your manuscript and increased the publishing potential of your work.

Best Regards

Comments on the Quality of English Language

Minor editing